# Benzodiazepine Modulation of GABA_A_ Receptors: A Mechanistic Perspective

**DOI:** 10.3390/biom12121784

**Published:** 2022-11-30

**Authors:** Marcel P. Goldschen-Ohm

**Affiliations:** Department of Neuroscience, University of Texas at Austin, Austin, TX 78712, USA; marcel.goldschen-ohm@austin.utexas.edu

**Keywords:** benzodiazepines, GABA_A_ receptor, mechanism, structure function

## Abstract

Benzodiazepines (BZDs) are a class of widely prescribed psychotropic drugs that target GABA_A_ receptors (GABA_A_Rs) to tune inhibitory synaptic signaling throughout the central nervous system. Despite knowing their molecular target for over 40 years, we still do not fully understand the mechanism of modulation at the level of the channel protein. Nonetheless, functional studies, together with recent cryo-EM structures of GABA_A_(α1)_2_(βX)_2_(γ2)_1_ receptors in complex with BZDs, provide a wealth of information to aid in addressing this gap in knowledge. Here, mechanistic interpretations of functional and structural evidence for the action of BZDs at GABA_A_(α1)_2_(βX)_2_(γ2)_1_ receptors are reviewed. The goal is not to describe each of the many studies that are relevant to this discussion nor to dissect in detail all the effects of individual mutations or perturbations but rather to highlight general mechanistic principles in the context of recent structural information.

## 1. Introduction

BZDs are one of the most frequently prescribed classes of psychotropic drugs today. In the United States, approximately 1 in every 25 adults was prescribed a BZD in 2021 [1]. Their sedative, hypnotic, anxiolytic, and anticonvulsant effects are used to treat a wide range of conditions, including anxiety, panic, insomnia, seizures, muscle spasms, and alcohol withdrawal [2]. Although largely successful, the prolonged use of BZDs can have undesirable side effects, including tolerance, dependence, withdrawal, and cognitive impairment [3,4]. Furthermore, BZDs are often present in drug-related overdoses, where they tend to be co-abused with alcohol and opioids [5,6]. Despite these side effects, their effectiveness as anxiolytics and muscle relaxants sees their continued frequent use as therapeutics.

BZDs are a class of molecules with a core consisting of fused benzene and diazepine rings. They target specific recognition sites in GABA_A_Rs, the primary inhibitory neurotransmitter receptor throughout the central nervous system [7]. The binding of the neurotransmitter GABA to GABA_A_Rs promotes the opening of the channel pore, allowing chloride influx into the cell, which typically inhibits neural signaling and is crucial for balancing excitatory and inhibitory signals during normal cognition. By modulating the response of GABA_A_Rs to GABA, BZDs tune GABAergic inhibition in the nervous system.

Different BZDs elicit different effects, which are broadly categorized as either enhancing (positive allosteric modulators, PAMs), inhibiting (negative allosteric modulators, NAMs), or having no effect (competitive antagonists) on GABA_A_R responses to GABA [8]. The sedative and anxiolytic effects of BZDs in clinical use reflect enhanced GABAergic inhibition by PAMs [2,9,10]. Here, if the subclass of BZD is not specified, a PAM should be assumed. Note that some BZDs act as either a PAM or a NAM depending on the GABA_A_R subtype. This article focuses on GABA_A_(α1)_2_(βX)_2_(γ2)_1_ receptors for which high-resolution structures are available in complex with BZDs.

## 2. A Brief History

The first BZD, chlordiazepoxide (Librium), was synthesized in 1955 by Leo Sternbach while working at Hoffmann-La Roche [11]. The discovery of the sedative and anticonvulsant effects of chlordiazepoxide, and soon thereafter the BZD diazepam (Valium), preceded knowledge of their molecular target [12,13]. Both chlordiazepoxide and diazepam were approved by the FDA in the early 1960′s and were rapidly adopted as a replacement to barbiturates, which have more severe side effects. They were originally referred to as tranquilizers, although terms such as anxiolytics and anticonvulsants are favored today.

Despite their widespread use, it was not until 1977 that an analysis of radiolabeled [^3^H]diazepam binding to brain tissue identified a specific receptor for BZDs localized to the central nervous system [14,15,16,17]. Observations that the binding potency of GABA analogues correlates with their ability to reversibly alter the affinity of BZDs for their receptor suggested that the BZD receptor may be part of a complex with GABA_A_R [18]. Using BZD-affinity chromatography, the BZD receptor was isolated from brain tissue, and its association with a GABA_A_R was confirmed by its high-affinity binding of both BZD and a GABA_A_R agonist [19,20,21,22]. An analysis of the purified receptor showed that it was heteromeric and composed of at least two different subunits [23]. Peptide sequences from the purified receptor were used to screen DNA libraries and identify the DNA sequences for the GABA_A_R α and β subunits, which formed functional GABA_A_Rs when coexpressed in Xenopus oocytes [24,25,26]. A third subunit, γ, was cloned later and is required to be coexpressed along with the α and β subunits for BZD binding and sensitivity [27].

Thereafter, multiple variants of the α (1–6), β (1–3) and γ (1–3) subunits were discovered [28], whose combinations confer differential effects of BZDs [29,30,31,32,33,34,35,36,37,38,39,40,41]. The cloning of GABA_A_R subunits enabled the molecular mechanism of BZD modulation to be probed via mutagenesis, which has led to numerous studies that identified not only the binding site but also the regions contributing to the drug’s modulatory effect. Advances in cryo-electron microscopy (cryo-EM) have recently enabled the resolution of structures of heteromeric GABA_A_(α1)_2_(βX)_2_(γ2)_1_ receptors in complex with BZDs [42,43,44,45,46,47,48]. Together, these observations aid in the continued search for a complete molecular understanding of the mechanisms of action of BZDs on GABA_A_Rs.

## 3. Canonical Extracellular High-Affinity Binding Site

The most prominent GABA_A_R subtype at synapses is (α1)_2_(βX)_2_(γ2)_1_ [49]. Notably, α4- and α6-subunit-containing receptors are insensitive to BZDs [38]. A comparison of α1 with α6 revealed a single histidine residue found in α1–3 and 5 that is essential for high-affinity BZD binding and is an arginine in α4 and 6 [50]. Swapping this residue either confers (histidine) or abolishes (arginine) high-affinity BZD PAM binding for both the α1 and α6 subunits [50,51]. Knock-in mice with this histidine to arginine mutation in α1 are insensitive to the sedative effects of diazepam, although they exhibit other anxiolytic and motor-impairing effects, possibly via receptors with α2, 3, or 5 subunits [29,30,31,52].

Mutagenesis and photoaffinity labeling identified a region in the extracellular domain (ECD) at the α+/γ− subunit interface as the location of the high-affinity BZD binding site where mutations often impaired BZD binding or the modulation of GABA-evoked responses [34,53,54,55,56,57,58,59] (Figure 1A,B). Cryo-EM structures confirmed this as a recognition site for BZDs [42,44,45,46,47,48]. The site is homologous to the two GABA binding sites at the β+/α− subunit interfaces, with the ligand binding in a pocket behind loop C, where it interacts with residues from both subunits (Figure 2). Rings in the BZD core structure form π-stacking interactions with surrounding aromatics. The critical histidine residue discussed above lies towards the back of the pocket, where it forms a hydrogen bond with the chlorine atom in PAMs such as diazepam and alprazolam (Xanax), the fluorine atom in the antagonist flumazenil, or the twin methoxy groups of the NAM DMCM (Figure 2*i*). Z-drugs such as zolpidem (Ambien), with similar sedative and hypnotic effects as BZDs, also bind at this site [48,60].

## 4. Transmembrane Binding Sites

The BZD diazepam (DZ) exhibits a biphasic concentration–response relation indicative of both high nanomolar affinity and lower micromolar affinity sites of action (Figure 1D). Whereas the mutations in the canonical ECD site discussed above are typically associated with the high-affinity response, residues in the transmembrane domain (TMD) have been similarly shown to be critical for the lower-affinity response [61]. Furthermore, mutations in the TMD of ρ_1_ subunit homomeric GABA_A_Rs that lack a BZD site can confer BZD sensitivity in the micromolar range, similar to the low-affinity response for heteromeric (α1)_2_(βX)_2_(γ2)_1_ receptors. Cryo-EM structural maps of GABA_A_(α1)_2_(βX)_2_(γ2)_1_ receptors in high micromolar concentrations of DZ show DZ bound at several of the TMD interfaces between neighboring subunits [45,46] (Figure 1C). Interestingly, these transmembrane (TM) sites overlap with sites for anesthetics such as propofol [62]. Mutations of several key residues in these sites, which are known to be important for anesthetic binding, impair the low-affinity action of BZDs such as DZ and midazolam [63].

It is important to note that biphasic concentration–response relations, as described for DZ, and binding to anesthetic sites in the TMD are not universal features of all BZDs. For example, structures in complex with alprazolam or flumazenil do not show binding in the TMD [45,46]. The effects of flurazepam are insensitive to mutations in these TMD sites [64], and flurazepam potentiation exhibits a bell-shaped concentration–response relation [65]. Nonetheless, the relevance of anesthetic sites of action in the TMD has been shown for BZDs such as DZ and midazolam [63] and remains to be tested for many other BZDs.

## 5. Other Binding Sites

In addition to the canonical ECD site at the α+/γ− interface, DZ also binds to a homologous site at the β+/γ− subunit interface in receptors lacking an α subunit, although such a receptor isoform may not contribute to behavior in neurons [66]. The compound CGS 9895, acting at the canonical α+/γ− site, has also been shown to bind to the homologous α+/β− ECD interface in αβ receptors without a γ subunit [65,67]. The ability of these ligands to bind to other intersubunit interfaces expand the potential for novel subtype-specific modulators [68,69]. However, the importance of these sites as drug targets has yet to be fully explored.

## 6. BZD Modulation of GABA Binding

BZD binding to the classical ECD site does not, by itself, result in any appreciable opening of closed channels but rather modulates the ability of agonists such as GABA to evoke pore opening. The enhanced GABAergic inhibition conferred by BZD PAMs largely stems from their potentiation and prolongation of GABA_A_R postsynaptic currents [2,9,10]. Strikingly, potentiation of current responses by BZDs occurs only for activation by subsaturating but not saturating concentrations of GABA, resulting in a leftward shift of the GABA concentration–response relation. This suggests that the mechanism underlying this potentiation involves enhancement of GABA binding [70,71,72,73,74]. At lower subsaturating GABA concentrations, where not all sites are occupied, enhanced GABA binding effectively results in larger current responses equivalent to those elicited by a higher concentration of GABA. In contrast, at saturating GABA concentrations, where all sites are already occupied, enhanced binding has no further effect. Slower GABA unbinding due to the stabilization of the bound complex can also explain the prolongation of the current decay by BZDs. Consistent with this idea, DZ has been shown to primarily affect the frequency of channel opening, which depends on the rate of GABA binding, whereas DZ has little effect on the duration of each opening event, which is governed by the energetics of pore closure downstream from the initial binding events (but see [72] as well as [70,75]).

For an effect on GABA binding, crosstalk between GABA and BZD sites is required. The enhanced binding of radiolabeled DZ in the presence of GABA in brain membranes supports the idea that agonists and BZD PAMs are costabilizing [76,77]. Observations of the rate of modification by methanethiosulfonate (MTS) reagents of introduced cysteines in GABA or BZD ECD sites indicate that binding at either the GABA or BZD sites induces distinct structural changes at the other site [78,79]. The dependence of these changes on the nature of the bound ligand (agonist or PAM, antagonist, or inverse agonist or NAM) suggests that these motions are relevant to the action of the ligand and indicate that GABA increases the aqueous accessibility to the BZD binding pocket.

For all ligand-gated ion channels, ligands that open the channel (agonists) bind more tightly to open (conducting) versus closed (non-conducting) channel conformations. Thus, any perturbation (e.g., BZD binding) that increases the probability of a channel to adopt an open conformation will necessarily increase the receptor’s apparent affinity for GABA, even if it has no effect on the affinities for each distinct closed and open state. Indeed, this can severely complicate the interpretation of equilibrium binding measurements [80,81]. Although BZDs clearly induce shifts in the apparent affinity for GABA, it is not yet fully understood to what degree this reflects direct changes in binding affinities vs. shifts in the closed–open equilibrium.

## 7. BZD Modulation of Pore Gating

Although some single-channel observations are consistent with the idea that BZDs primarily modulate GABA binding [72], other studies observed changes in macroscopic desensitization or single-channel open durations that are inconsistent with effects solely on agonist binding [75,82,83]. In addition, two lines of evidence strongly indicate that BZDs alter channel gating steps downstream of GABA binding. First, peak current responses to saturating concentrations of partial agonists are potentiated by BZDs [84,85,86,87]. Partial agonists bind at the GABA sites but elicit smaller current responses than full agonists such as GABA. Nonetheless, at saturating concentrations the sites are fully occupied, and thus the increased maximal current response conferred by BZDs must reflect the modulation of gating conformational changes in the partial agonist-bound receptor, regardless of any effects on the binding step itself.

Second, BZDs directly gate mutant receptors in the absence of GABA [84,85,86,87,88]. A mutation in the pore gate confers a channel that spontaneously exchanges between closed and open conformations [89,90]. The closed–open equilibrium of such spontaneously active mutants is modulated by BZDs alone, an effect that necessarily does not involve the binding of agonists such as GABA. Although these observations require mutating the receptor in such a way as to significantly alter the resting equilibrium, it is likely that the mutant still gates by the same mechanism as wild-type channels. In support of this idea, a single-channel analysis of mutants and combinations thereof that confer such spontaneous unliganded gating in homologous nicotinic acetylcholine receptors (nAChRs) showed that these mutants alter the closed–open equilibrium independent of agonist-elicited gating [91]. This suggests that the chemical energy from BZD binding does affect the pore gate, but the amount of energy is insufficient to elicit appreciable channel opening without additional energy from either agonist binding or another perturbation, such as a mutation.

## 8. BZD Modulation of Closed-Channel Pre-Activation

Single-channel gating dynamics for full versus partial agonists at homologous nAChR and glycine receptors (GlyRs) can be rationalized by a mechanism whereby agonist binding is associated with the receptor adopting a “flipped” or “primed” pre-active conformation that precedes pore opening [92,93,94,95]. This model discriminates between partial and full agonists based on their ability to bias the receptor towards this pre-active intermediate state, following which pore opening/closing is ligand-independent.

Strikingly, if one assumes that BZDs influence a similar “flipping” or “priming” pre-activation step in GABA_A_Rs [96], apparently conflicting observations supporting BZD-modulation of either agonist affinity or pore gating can both be readily explained [97,98,99]. By modulating the probability of receptor pre-activation, BZDs regulate the amount of time spent in a state from which the channel can open. Under conditions where pre-activation is not highly probable, enhanced pre-activation by BZDs will increase the frequency of channel opening. This explains the observed BZD-potentiation of currents evoked with either partial agonists or subsaturating GABA concentrations where only one of the two agonist sites are occupied. Conversely, in saturating GABA with both sites occupied, the channel is already so strongly biased towards its pre-active state that a further enhancement of pre-activation by BZDs has little to no additional effect on channel opening. This explains the lack of potentiation of currents evoked by saturating GABA. A similar mechanism can explain the modulation by non-BZDs such as the Z-drug zolpidem, which binds at the canonical BZD site [100].

Given its explanatory power, it is highly plausible that such a mechanism underlies many of the observed effects of BZD binding in the ECD. Nonetheless, direct evidence for this mechanism in GABA_A_Rs is lacking. In GlyRs and nAChRs, the pre-active state preceding pore opening is very short-lived, and a similarly transient non-conducting intermediate in GABA_A_Rs challenges both functional and structural observations. Moreover, observations that single-channel open durations are altered by partial versus full agonists or BZDs are not explained by changes in pre-activation alone, which only affect the opening frequency [75,101]. Some of these complications may reflect BZD binding to TMD sites, which based on their proximity, could directly affect the pore gate. Otherwise, the totality of the effects of BZDs at the canonical ECD site is at least a bit more complex than the satisfyingly simple approximation that they modulate channel pre-activation.

## 9. Structural Mechanism for BZD Modulation in the ECD

Advances in cryo-EM have begun to provide structural snapshots of heteromeric GABA_A_(α1)_2_(βX)_2_(γ2)_1_ receptors in complex with both agonists and BZDs [42,43,44,45,46,47,48]. These studies validate the canonical high-affinity BZD site in the ECD, as identified by functional studies, and reveal details as well as subunit specificity of lower-affinity BZD sites in the TMD (Figure 1A–C). Together with structures of homologous pentameric ligand-gated ion channels (pLGICs) such as nAChRs, GlyRs, 5-HT_3_ receptors, and prokaryotic homologues [102], these structural snapshots are generally in agreement with the overall gross subunit motions occurring during agonist gating. A comparison of agonist-bound structures with either unliganded or antagonist-bound structures indicate that agonist binding is associated with the ECDs adopting a more compact conformation with an increase in the buried surface area at the intersubunit interfaces and loop C moving inward over the agonist. In addition, the ECD of each subunit undergoes a rotation and tilt that results in an expansion at the ECD/TMD interface, where the M2-M3 linkers move radially outward and the top of the pore-lining M2 helices splay apart, resulting in the opening of the hydrophobic 9′ leucine ring near their center, which constitutes the activation gate.

Comparisons of the structures of GABA_A_(α1)_2_(β2–3)_2_(γ2)_1_ receptors with and without classical BZDs such as DZ show, at most, only subtle conformational differences in the ECD [44,45,46,103]. Based on this observation, Masiulis et al. suggested that BZD PAMs act at their high-affinity site to stabilize the α+/γ− subunit interface, which they proposed should aid in the global compaction of the ECD during gating and thereby promote channel opening [45]. In contrast to PAMs, occupation of the α+/γ− ECD site by the BZD antagonist flumazenil confers a slight expansion in the ECD [103].

Consistent with the idea that the stability of the α+/γ− subunit interface is important for BZD modulation, cysteine cross-linking between α and γ subunits in close proximity to the critical histidine residue at the back of the canonical BZD site in the α subunit β4–β5 linker mimics the action of BZDs and is reversible upon breaking the disulfide bond [104] (Figure 2*ii*). In contrast, the insertions of glycine residues to increase the flexibility of β4–β5 linkers in the α, β, or γ subunits all impair the BZD potentiation of GABA-evoked currents [105] (Figure 2*iii*). The importance of the linker in all subunits suggests that BZDs are associated with global changes that could couple BZD and agonist binding sites via interactions between the β4–β5 linker at the back of each binding site and the neighboring subunit beta barrel. Indeed, mutations affecting BZD modulation are not localized to a specific domain, supporting the notion of a more global effect on the channel conformation.

Given the rigid body movement of the subunit ECD beta barrels, it is natural to expect much of the dynamics to occur at the interfaces between subunits, which include the GABA and BZD binding sites [106]. Only one of the two GABA sites shares the same α subunit as the BZD site, raising the possibility that BZDs may induce local changes that preferentially modulate activation via a single GABA site. However, the serial disruption of each of the two GABA binding sites in concatenated receptors revealed that BZDs can modulate currents in response to GABA binding at either site [107]. Apart from BZD- and agonist-binding α+/γ− and β+/α− interfaces, respectively, epilepsy-related mutations that disrupt electrostatic interactions at the canonical nonbinding γ+/β− interface reduce the stability of the BZD-receptor complex [108]. Thus, the destabilization (or stabilization) of one interface results in a more global destabilization (or stabilization) at other interfaces, consistent with the idea that BZDs influence the stability of intersubunit contacts throughout the ECD.

Although it is likely that BZDs stabilize the α+/γ− interface, it is important to note that all structures of BZD-bound GABA_A_(α1)_2_(β2–3)_2_(γ2)_1_ receptors to date were obtained with both agonist sites occupied by GABA. Indeed, the inability of BZDs to potentiate currents evoked by saturating GABA is hypothesized to reflect a nearly complete biasing of the receptor to a pre-active conformation by the relatively larger energetic contribution from both bound agonists that effectively overshadows the much smaller contribution from BZD binding in the ECD. Thus, it is perhaps not surprising that little to no BZD-associated motions of the ECD were detected in this condition. Structures of BZD complexes with zero or one bound agonist or with partial agonists that may better inform on the motions associated with BZD binding are currently lacking.

Structural models of homologous GlyR and ACh binding protein in complex with partial agonists suggest that the initial pre-activation step involves loop C swinging towards the bound agonist to adopt a more compact conformation [109,110]. Although the styrene maleic acid (SMA) copolymer used to solubilize GlyR in this study likely biased the conformations to some extent, this bias may have serendipitously aided the observation of a normally transient pre-active state. Taken together with the structures of GABA_A_(α1)_2_(βX)_2_(γ2)_1_ receptors, this suggests a hypothetical mechanism for BZD modulation via the canonical high-affinity site: BZD PAMs promote the adoption of a pre-activated more compact ECD conformation with loop C closed more tightly around the binding pocket by stabilizing or increasing global intersubunit interactions. It is straightforward to assume that NAMs might have an opposite effect by destabilizing intersubunit interfaces. Consistent with this idea, GABA_A_(α1)_2_(β2)_2_(γ2)_1_ receptor structures in complex with either a BZD antagonist or NAM exhibit a less compact ECD than complexes with PAMs [46,48].

## 10. Structural Mechanism for BZD Modulation in the TMD

To ensure the saturation of the binding sites, GABA_A_(α1)_2_(βX)_2_(γ2)_1_ structures were solved in high micromolar concentrations of DZ sufficient to populate lower-affinity sites in the TMD. The observed TMD binding sites validate binding residues identified by mutagenesis and highlight asymmetric binding to specific subunit interfaces [46,63]. DZ binds in an intersubunit pocket between the TM helices of the β and α subunits directly below the GABA binding sites as well as in the homologous pocket at the γ+/β− interface. The pore-lining M2 helices contribute to the binding pockets through which DZ may directly affect pore opening. Interestingly, these are the same TM sites that bind anesthetics such as propofol at the β+/α− interface and barbiturates such as phenobarbital at the γ+/β− interface [46,62,111]. Thus, lower-affinity binding to these sites may contribute to the anesthetic effects of high concentrations of BZDs such as DZ. At nanomolar to low micromolar concentrations of DZ, as explored by many functional studies, the observed effects are likely to be primarily due to binding at the canonical high-affinity ECD site. Nonetheless, in the absence of evidence to the contrary, binding at the TM sites in a fraction of receptors could complicate the interpretation of these data.

In contrast to the relative lack of distinct BZD-associated motions in the ECD, BZDs have more clear effects on the structure of the TMD. Diazepam binding in the TMD results in a global stabilization of the TMD, with tighter packing of the TM helices from neighboring subunits. Strikingly, although the BZD antagonist flumazenil does not bind in the TMD, flumazenil binding to the canonical ECD site destabilizes the TMD and increases the gap between neighboring subunit TM helices. This suggests that BZD binding to the canonical ECD site can induce structural changes in the TMD. Consistent with this idea, the accessibility of TM domains to MTS reagents is altered by BZDs, an effect that is asymmetric amongst subunits [112,113].

Although BZD modulation of TMD stability was observed by one group [46,103], another group observed very little change in the TMD [44,45]. One explanation for this discrepancy may be that the tighter wrapping of the nanodisc scaffold in the latter group’s preparation could have constrained TMD motions and prevented BZD-induced changes. Alternatively, the truncation of the intracellular M3-M4 loop by the former group may have resulted in an abnormally destabilized TMD. Both groups used nanobodies bound to the ECD to aid in determining molecule orientation, which could affect ECD conformations. The degree to which the preparations influenced the resulting structures is a very important question for interpreting the structural mechanisms that remain to be fully understood.

## 11. BZD-to-Pore Coupling

Although the general idea that BZDs such as DZ stabilize a more compact pre-active conformation of the receptor, with intersubunit interfaces adopting more extensive interactions, is supported by a variety of experimental observations, the specific interactions and microdomain motions that are crucial for BZD modulation remain to be fully elucidated. For example, cysteine cross-linking between the γ subunit β8–β9 loop beneath the BZD binding site and either the neighboring β9 strand in the same subunit or the β1–β2 loop in the neighboring α subunit impair modulation by the BZD flurazepam, suggesting that a more flexible β8–β9 loop is important for the modulatory effects of BZDs [114] (Figure 2*iv*–*vi*). In contrast, cross-linking the β9 loop to the neighboring pre-M1 region that connects the ECD to the M1 helix enhances flurazepam modulation (Figure 2*vii*). The γ subunit β8-β9 loop extends along the bottom outer edge of the BZD binding pocket to the intersubunit region near the ECD/TMD interface, where it can interact with the M2-M3 linker, Cys-loop, and β1–β2 loop in the neighboring α subunit, a region known to be important in channel activation [115,116] (Figure 2). This loop has been implicated in both the transduction of the modulatory effects and the binding of BZDs [57,60,117,118,119]. The homologous loop in the α subunit has also been implicated in gating by GABA [120] and was observed to undergo relatively large motions upon the activation of the prokaryotic homologue GLIC [121].

One complication in the interpretation of studies that examine BZD modulation of GABA-evoked currents is the ambiguity in attributing observed effects to either a direct change in pore gating or an indirect change in the pore closed–open equilibrium arising from a direct change in GABA affinity. A method of disambiguating between these effects is to specifically measure the energetic perturbation that BZD binding confers to the pore gate in the absence of GABA. Spontaneously active mutants whose closed–open equilibrium is sensitive to BZDs alone enable such an approach. Using this strategy, a single valine residue in the center of the α1 subunit M2-M3 linker was identified that when mutated to alanine confers an approximately three-fold increase in the efficiency of the transduction of chemical energy from DZ binding to the pore gate [88] (Figure 2*viii*). The reduction in sidechain volume for the alanine substitution may contribute to this effect, as a mutation to a bulkier tryptophan did not enhance DZ efficiency, suggesting that a more flexible M2-M3 linker may also promote BZD modulation, possibly via interactions with the neighboring γ subunit pre-M1 region or β8-β9 loop. How these observations relate to a more globally compact receptor remains to be fully understood, as does the symmetry of the mechanism for PAMs and NAMs for which perturbations do not always exhibit mirrored effects [114].

## 12. Conclusions and Perspectives

Despite considerable interest in uncovering the basis for BZD modulation, we still do not fully understand how these drugs work at the molecular level. Although numerous studies have shed light on the underlying mechanism and a series of observations have highlighted a role in aiding receptor pre-activation, major questions remain. From a structural perspective, three-dimensional information about BZD-bound receptor conformations in conditions where BZDs are known to have functional effects (e.g., in complex with partial agonists or a single bound agonist) are lacking. For BZDs such as DZ that bind in both the ECD and TMD, structures with only one of the domains occupied would aid in dissecting the distinct effects of binding in each domain. More generally, a structure of a heteromeric GABA_A_R with a conducting pore is lacking, as all agonist-bound structures are in a putative desensitized state with a constriction at the bottom of the M2 pore-lining helices [122,123]. Obtaining these structures likely represents a significant challenge due to the transient nature of the pre-active and open conformations, which results in agonist-bound receptors rapidly accumulating in desensitized states. Furthermore, a better understanding of the degree to which solubilization scaffolds or nanobodies influence channel conformation is also imperative to reliably interpret these observations. Functionally, much work remains to be carried out to dissect the detailed molecular interactions that are involved in BZD modulation. Use of gain-of-function mutants that enable the unambiguous measurement of BZD-to-pore coupling energetics without the complication of agonist binding may prove fruitful in this regard. Thus, although a general understanding of the mechanism of BZD modulation is beginning to emerge, there is much that remains to be discovered to fully elucidate the physical basis for this important class of psychotropic modulators.

## Figures and Tables

**Figure 1 biomolecules-12-01784-f001:**
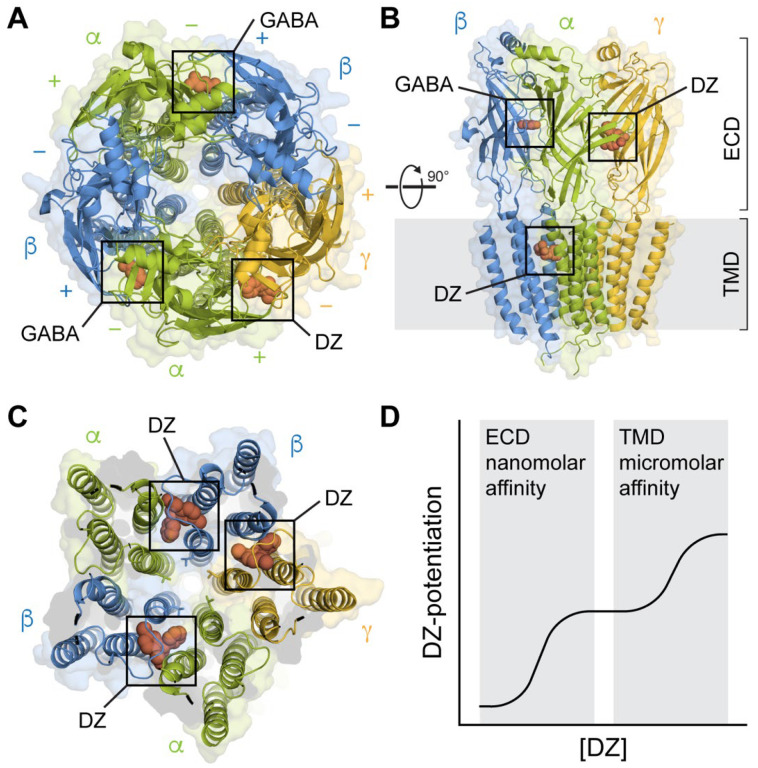
Structures of synaptic GABA_A_Rs in complex with the BZD PAM diazepam (DZ) at both high-affinity ECD and lower-affinity TMD sites. (**A**) Top-down view with GABA bound at the β+/α− interfaces and DZ bound at the α+/γ− interface in the ECD. Cryo-EM map of GABA_A_(α1)_2_(β3)_2_(γ2)_1_ from PDB 6HUP. (**B**) Side view of the structure in A, additionally showing one of several binding sites for DZ in the TMD. Only three subunits are shown for clarity. (**C**) Same perspective as in (**A**) for a slice through the TMD. Cryo-EM map of GABA_A_(α1)_2_(β2)_2_(γ2)_1_ from PDB 6X3X. Three binding sites for DZ are highlighted at the β+/α− and γ+/β− intersubunit pockets between the TM helices and below the M2-M3 linker of one of the subunits. The central pore-gate-forming 9′ leucine residues are shown as sticks near the middle of the pore-lining M2 helices. (**D**) Biphasic modulation by DZ at the canonical high-affinity site in the ECD and lower-affinity sites in the TMD.

**Figure 2 biomolecules-12-01784-f002:**
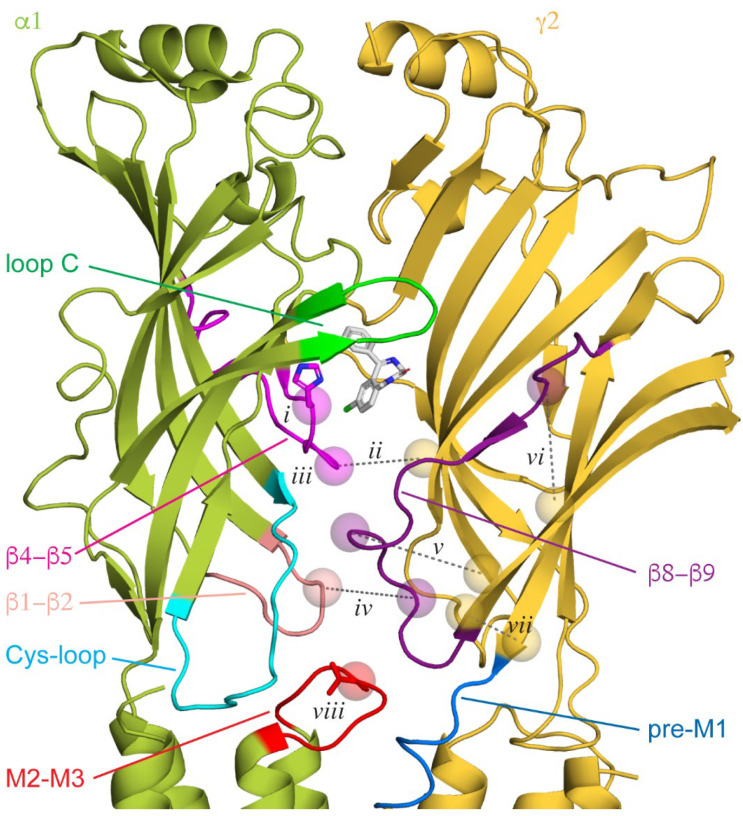
Canonical ECD binding α+/γ− interface, with diazepam (DZ) shown as white sticks bound behind loop C. Other loops implicated in binding or intersubunit interactions are indicated. Cryo-EM map for GABA_A_(α1)_2_(β3)_2_(γ2)_1_ from PDB 6HUP. Features discussed in the main text are indicated with spheres for residue Cα atoms and dashed lines for cross-linked residue pairs (numbered labels are referenced in the main text). The critical histidine residue in the β4−β5 loop within the BZD binding pocket (*i*) and the central valine residue in the α1 M2-M3 linker (*viii*) are shown as sticks.

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
