# Peer review of "Benzodiazepine Modulation of GABAA Receptors: A Mechanistic Perspective"

_biomolecules, 2022, doi:10.3390/biom12121784_

Round 1
Reviewer 1 Report
This review is a nice piece of work in which current structural knowledge, mutational/ functional data, and mechanistic models for benzodiazepine modulation of GABA-A receptors are presented and discussed.
Prior to publication, some points should be addressed as follows:
Nomeclature: Subunit isoforms are subscripted, which is against current IUPHAR recommendation. It is strongly recommended to adhere to IUPHAR standards throughout the review.
Line 59: The sentence should be reworded, variants of a1-6 etc. does not make sense.
Lines 97 and 100: It is not correct to generalize the biphasic/ bisigmoid conc.-response curves to BZDs: For flurazepam it is known to be bell shaped, some BZDs don't have any second phase, and for most it is unknown, as such curves are published only for a few benzodiazepines. In the schematic panel D of Fig 1, DZ is correctly specified, this should be also in the figure legend and in the text to make clear that the case of diazepam is discussed. (A list of additional KNOWN examples would be helpful, and perhaps explicit mentioning of the bell shaped flurazepam response (doi: 10.1111/j.1471-4159.2008.05574.x., cited as 67) would also be of interest.)
Lines 116-124: This short chapter on other binding sites should be improved to avoid unspecific statements ("one ecompound"), and to avoid invalid generalization (BZDs).
Line 236: "45 suggest" should be reworded to state "Masiulis et al. suggest"
Figure 2: The figure legend (lines 253-255) fails to inform about the depicted object (PDB ID or specification of an own model, whichever applies). The rendered His in loop A is not labelled. The figure is called multiple times to illustrate various findings, and fails to depict the details that would be needed: Lines 241-254, and lines 333-340 describe objects that could, and should be rendered and labeled in the Figure.
lines 300-302: Do all BZDs have anaesthetic effects? Do all of them bind to anaesthetics sites? The passage should be reworded to reflect the limited data for the majority of BZDs.
Line 317: It is incorrect to state that no structure exists with only the high affinity site ligated, there is an alprazolam bound structure - and the work is actually cited.
I hope the author finds the suggestions helpful to further improve this review.
Reviewer 2 Report
The review submitted by Marcel P. Goldschen-Ohm reports insights into the structural biology of GABAA receptor and is written by a scientist highly specialized on this topic. I have only minor points to be adressed or discussed by the author upon resubmission. English quality is below standards and awaits improvements.
Minor concerns.
1. First page of this submission (Abstract and Introduction) is written in awkward style and is misleading, speaking in general about the GABAA receptor. It should be emphasized already here that the review is devoted to one of the most studied of recombinant receptors type: a1bxg2. Otherwise it is unclear to which drugs NAM (negative allosteric modulator) is referred here: the gamma 1-containing GABAARs are potentiated by most of the "NAM" for gamma2-containing GABAA receptors (line 86). As the abbreviations PAM, NAM and FAM are not generally used in the GABAAR field I suggest to call the drugs by chemical names rather than by these abbreviations.
2. The word "synaptic" should be avoided in this manuscript when spaking about GABAAR. Why not to speak about "extrasynaptic" particularly regarding action of the most frequently used anesthetic midazolam (with the preference for extrasynaptic a5b3g2 receptor).
3. The word "in silico electrophysiology" should be introduced, as most suitable to the field described in this manuscript.
4. Concentration range of benzodiazepines interacting with extracellular loop of GABAAR vs transmembrane domain (Fig.1) should be clearly provided and compared with the clinically relevant doses.
5. When speaking about physiological actions of GABA and pharmacologically defined GABAAR receptor types, it would be nice to cite in the introduction one of the reviews from the same special issue. This citation should be suggested to the author by the editor of this special issue.
Reviewer 3 Report
In this review, the author masterfully takes a new approach to the interactions of benzodiazepines (BZDs), a class of widely prescribed psychotropic drugs that target GABAA receptors (GABAARs), in the context of recent structural information. The manuscript is well-written, and the different sections provide relevant information and a modern vision of the proposed topic. It would have been interesting if the author had also devoted a section to interpreting what effect the occurrence of mutations (missense mutations and variants) associated with epilepsy syndromes in locations close to the BZD binding sites could have. Since GABAARs are important targets in different epileptic phenotypes, a holistic view of how structural changes resulting from mutations at or near the BZD binding site can explain changes in receptor behavior toward BDZs. Therefore, opening possible interpretations for the mechanism of resistance to antiepileptic drugs (AEDs) in "refractory"/"intractable" epilepsies.
Author Response
I thank the reviewer for their kind words. I also agree that a discussion of the effects of mutations associated with epilepsies on BZDs would certainly be of interest. However, I feel it would be more deserving of its own review, or perhaps in conjunction with other observations related to human disease.